# Pan-Genomic Regulation of Gene Expression in Normal and Pathological Human Placentas

**DOI:** 10.3390/cells12040578

**Published:** 2023-02-10

**Authors:** Clara Apicella, Camino S. M. Ruano, Basky Thilaganathan, Asma Khalil, Veronica Giorgione, Géraldine Gascoin, Louis Marcellin, Cassandra Gaspar, Sébastien Jacques, Colin E. Murdoch, Francisco Miralles, Céline Méhats, Daniel Vaiman

**Affiliations:** 1Team ‘From Gametes to Birth’, Institut Cochin, U1016 INSERM, UMR 8104 CNRS, Paris-Descartes University, 75014 Paris, France; 2Fetal Medicine Unit, St George’s University Hospitals NHS Foundation Trust, London SW17 0RE, UK; 3Vascular Biology Research Centre, Molecular and Clinical Sciences Research Institute, St George’s University of London, London SW17 0RE, UK; 4Department of Neonatology, Angers University Hospital, F-49000 Angers, France; 5Department of Gynaecology, Obstetrics and Reproductive Medicine, Centre Hospitalier Universitaire (CHU) Cochin Faculté de Médecine, Assistance Publique-Hôpitaux de Paris (AP-HP), Hôpitaux Universitaires Paris Centre (HUPC), Université de Paris, 138 Boulevard de Port-Royal, 75014 Paris, France; 6Sorbonne Université, Inserm, UMS Production et Analyse des données en Sciences de la vie et en Santé, PASS, Plateforme Post-génomique de la Pitié-Salpêtrière, 75013 Paris, France; 7Systems Medicine, School of Medicine, University of Dundee, Dundee DD1 9SY, UK

**Keywords:** placenta, preeclampsia, expression Quantitative Trait Loci

## Abstract

In this study, we attempted to find genetic variants affecting gene expression (eQTL = expression Quantitative Trait Loci) in the human placenta in normal and pathological situations. The analysis of gene expression in placental diseases (Pre-eclampsia and Intra-Uterine Growth Restriction) is hindered by the fact that diseased placental tissue samples are generally taken at earlier gestations compared to control samples. The difference in gestational age is considered a major confounding factor in the transcriptome regulation of the placenta. To alleviate this significant problem, we propose here a novel approach to pinpoint disease-specific cis-eQTLs. By statistical correction for gestational age at sampling as well as other confounding/surrogate variables systematically searched and identified, we found 43 e-genes for which proximal SNPs influence expression level. Then, we performed the analysis again, removing the disease status from the covariates, and we identified 54 e-genes, 16 of which are identified de novo and, thus, possibly related to placental disease. We found a highly significant overlap with previous studies for the list of 43 e-genes, validating our methodology and findings. Among the 16 disease-specific e-genes, several are intrinsic to trophoblast biology and, therefore, constitute novel targets of interest to better characterize placental pathology and its varied clinical consequences. The approach that we used may also be applied to the study of other human diseases where confounding factors have hampered a better understanding of the pathology.

## 1. Introduction

Eutherian mammals made a drastic evolutionary ‘choice’ for survival, the one to host the embryo and to develop the fetus over a relatively extended period of time. Since the feto–placental unit is, in fact, a semi-allograft, that in all other circumstances should be rejected by the maternal immune system. This tolerance period ranges from a few days in marsupials (13 days in the Virginia opossum, where poor coping with inflammation apparently shortens the length of the feto-maternal tolerance period [1]) to 22 months in the African elephant. The pivotal organ for this heavy evolutionary option is the placenta, which manages immunotolerance, exchange of nutrients, withdrawal of waste, and pro-gestational hormone production. The inherent complexity associated with this role of the interface makes its regular function a fragile equilibrium, which in humans evolve relatively often towards placental diseases, such as Hypertensive Disorders of Pregnancy (HDP), Intra-uterine Growth restriction (IUGR), Gestational diabetes or other diseases related to abnormal implantation [2,3]. Despite this obvious fundamental role, understanding the genetic function of the human placenta is a difficult challenge, and overall this organ is not explored as much as other organs. For instance, the Genotype Tissue Expression (GTex) project, for instance, provides a comprehensive list of expression-Quantitative Trait Loci (eQTLs), as well as splicing-QTLs (sQTLs) in 49 human tissues, that do not include the placenta [4] Recent progress using global approaches made it possible to better analyze the regulation of gene expression in the pathological placenta, such as in preeclampsia and Intra-Uterine Growth Restriction [5,6]. Genetic regulation of gene expression by genetic variants has been systematically analyzed in three recent articles that initiated deciphering the landscape of genetically controlled genes in the placenta [7,8,9]. Our data presented herein revisit the question of eQTL in the placentas and uses a novel methodological approach to identify actual differences between normal and pathological placentas often blurred by a major confounding factor, such as placental age. For this, we gave a strong weight to the use and the definition of covariates. Covariates were defined by available clinical (i.e., age of the placenta, disease status) and technical parameters (i.e., sample preparation, origins), by principal components discovered from the transcriptome analysis, and through the systematic identification of Surrogate Variables [10].

Since confounding factors are a major issue in placental genetics (especially the gestational age, the mode of delivery–C-section or Natural), we develop a novel approach that could be used to identify relevant disease-associated genetic variants (SNPs) associated with alterations of gene expression. In the first step, the QTL analysis was performed considering the disease status as a covariate, and then, in the second step, we did not correct for disease status looking for eQTLs that might be influenced by genetics and disease and being, therefore, potentially interesting in the context of placental pathologies. We surmise that the additional couple of SNP-gene found were mostly and uniquely specific to the disease. While it is not possible to affirm that the gene expression alterations are causal to the disease, they could later be evaluated in other human cohorts of placental DNAs to assess their potential predictive value heuristically.

The approach is summarized in Figure 1. This alternative analysis strategy could be applicable to a wide range of eQTL studies, especially those in which gene expression is believed to be highly influenced by clinical and technical variables and that is limited by small sample sizes, such as those including disease samples.

## 2. Methods

### 2.1. Summary of the Principle

In the first step, we performed the analysis including the disease status (Control, Pre-eclampsia (PE), IUGR, and PE + IUGR) as a covariate, which allowed us to identify eQTLs that influence placental gene expression independently of the disease. In order to restrict the number of input genes for the analysis, we first built a linear model using the gene expression levels for each gene in the function of the covariates, then we selected a subset of potentially relevant genes based on their residual variance (the variability of their expression level) since genes for which there is no residual variance once the covariate effects are removed are not going to reveal any detectable effect that could be partly explained by changes at the genetic level. This approach has similarly been used in a QTL research paper for genomic methylation QTL (meQTL, [11]).

This step of identification of eQTLs confirmed a large part of previously identified couples eSNP-eGenes discovered in previous studies and identified new ones, presenting a strategy that can be applied with a limited sample size (<100). 

In a second step, we aimed to identify genes that are modified during complicated pregnancy by the disease specifically and which present a component of genetic regulation. Thus, we removed the disease status from the group of covariates and submitted the data for analysis again in order to identify a novel list of eQTLs. This approach led to the discovery of 16 cis-eQTLs that are potentially associated with placental diseases. We surmise that these genetic factors identified may have been overlooked in previous analyses since they could have largely been confounded with the covariates. In particular, in our approach, we managed to partially account for critical confounding covariates such as placental age, which is often difficult to dissociate from placental disease. Interestingly, both approaches identified genes where we could observe statistical associations between disease status and genotype at the locus.

### 2.2. Human Placental Samples

This study included the use of human placental samples from three different cohorts for a total of 66 samples. For all the cohorts, the participant gave their informed consent. For the St George’s cohort, the administrative ethical references were 19/LO/0974, approved by the Brent Research Ethic Committee (London) on 28 June 2019. For the Angers cohort, the study was approved by the Ethics Committee of Angers. The cohort was registered at the French CNIL (Commission Nationale de l’Informatique et des Libertés no. pWP03752UL, ethics committee for the collection of clinical data from patient records). The study was validated by the French CPP (Comité de Protection des Personnes) and registered to the French Ministry of Research under number DC-2009-907. Finally, for the Co-chin hospital cohort, the ethical administration was given by the CPP ‘Ile de France XI’ under the reference number 11018, 3 March 2011. 

Angers University Hospital and Cochin Hospital cohorts have been described in [12]. They included 8 control (CTRL, placentas obtained from uncomplicated pregnancies) and 13 IUGR samples from Angers, 9 CTRL, 7 PE, and 3 PE+IUGR from Cochin. Preeclampsia was defined by the presence of hypertension (systolic pressure > 140 mm Hg, diastolic pressure > 90 mm Hg) and proteinuria (>0.3 g/day). IUGR was defined as a reduction of fetal growth during gestation accompanied by Doppler abnormalities and birth weight below the 10th percentile [13,14]. The Gynecochin cohort samples were kindly shared by Dr. Louis Marcellin, and included 20 CTRL, 13 from natural delivery, and 7 from C-section. For placental sample collection relative to Angers, Cochin, and Gynecochin cohorts, sections of 1 cm3 of placental villi were dissected from four different cotyledons between the basal and chorionic plates, as previously described in [14]. St. George’s Hospital samples, 6 PE placentas from C-sections, were kindly provided by Prof. Basky Thilaganathan and MD Veronica Giorgione in the context of the iPlacenta consortium. PE was defined by de presence of hypertension according to guidelines by The International Society for the Study of Hypertension in Pregnancy [15]. Placental samples were collected from the middle region of the placenta containing only villous tissue, just above the cord insertion point, then washed in PBS. 

### 2.3. RNA and DNA Extraction

Human placental tissues were obtained from three different cotyledons of the maternal side of the placenta, washed extensively in sterile PBS, and snap frozen or in RNAlater (Invitrogen, Waltham, MA, USA), then powdered using pestle and mortar on dry ice or the hammering method in liquid nitrogen. The powdered tissues were then processed for DNA and RNA extraction. RNA was extracted with TRIzol (Invitrogen), according to manufacturer’s instructions; UltraPure™ Phenol:Chloroform:Isoamyl Alcohol (25:24:1, *v*/*v*) (Invitrogen) was used in place of Chloroform. Extracted RNA was resuspended in RNAse and DNAse-free water, and integrity was assessed with Bioanalyzer Agilent 2100 nano kit. For DNA extraction, powdered tissue was resuspended in Sample Lysis Buffer (10 mM Tris Hcl pH8, 10 mM EDTA, 50 mM NaCl, 0.5% SDS, Proteinase K (Invitrogen #25530-049) used at 1:3000) and incubated overnight in shaking water bath at 58 °C. Addition of absolute ethanol allowed precipitation of the DNA in filamentous, visible form, which was then transferred to new 1.5 mL tubes for additional washes in ethanol 70%. The pellet was left to air-dry, and DNA was resuspended in Tris-EDTA.

### 2.4. Transcriptomic Dataset

100 ng of RNA from the human placental samples were analyzed by ClariomD arrays (Applied Biosystems™, Affymetrix, Thermo Fisher Scientific, Montigny-le-Bretonneux, France) as described in [12]. This array measures gene expression both at the gene level as well as the exon level, providing also splicing isoform-specific data. The 66 samples were processed in three different batches, generating three sets of raw data files (.CEL). ***Merging datasets.*** To reduce batch effects in fluorescence signal due to experimental variability, the raw data (.CEL) files were processed together using the Transcriptome Analysis Console (TAC) software (Thermo Fisher Scientific), performing the default Robust Multi-Array normalization (RMA). This algorithm performs background adjustment, followed by quantile normalization and summarization [16]. The ClariomD array measures expression levels for a total of 134,748 probes, corresponding to coding and non-coding genes, 18,858 and 66,845, respectively, as well as predicted genes, pseudogenes, and small RNAs. Probe coordinates refer to the GRCh38 Genome Build. For downstream analyses, only genes with known GeneID and Description in TAC have been kept, removing transcripts identified by the database AceView, as well as transcripts on the Y and mitochondrial chromosomes; for a final total of 46,624 probes. Probes have been further filtered by mean fluorescent value (LOG2) across all samples ≥ 4.5, reducing the total number of probes to 33,988 considered to be “expressed”. ***Transcriptomic covariates.*** Classical clinical and experimental variables in placental studies were used for the whole sample and included a total of 9 variables (Batch, Cohort, Group, Delivery, Maternal Age, Ethnicity, Gestational Age, Sex, and Parity). In particular, Batch refers to the experimental batch of transcriptomic data acquisition, Cohort to the different cohort of origins of the placental samples relative to the Hospital in which they had been collected, Group to the disease status, Delivery to the delivery mode, either C-section or Natural delivery, and finally Sex to the placental sex. For these 9 variables, missing values were replaced with the mean of each disease group, according to the disease group of the sample with missing value [17]. Summary statistics by disease group are listed in Table 1, the complete dataset is available in Appendix A.

Principal Component Analysis (PCA) was performed in R (version 4.1.0, 2021-05-18) with the R package PCATools, on the 57 samples, for the transcriptomic dataset of 46,624 probes. The eigencorplot function within the package was used to calculate correlation coefficients between the first 10 principal components (PCs) and the known covariates; it performs a Pearson correlation, followed by F-statistic [18]. The R package SVA was used to identify potential additional sources of variation in the dataset, i.e., surrogate variables, by building the mod linear distribution with “Group” as the independent variable and setting the mod0, to the intercept [10]. For more details, see [12]. A correlation matrix was calculated in R, using as input the 9 clinical and experimental variables, the first 10 PCs of gene expression PCA, and 4 identified surrogate variables to identify the final set of variables correlated with global transcriptomic changes. A correlation coefficient cut-off threshold of |0.9| was used to remove colinear variables [19]. The final set of transcriptomic covariates includes Batch, Group, Delivery, Maternal Age, Ethnicity, Gestational Age, Sex, Parity, and Transcriptome PC1 to PC5.

### 2.5. Genotype Dataset

For the 66 samples, 200 ng of genomic DNA were genotyped using the Infinium OmniExpress (illumina) BeadChip, which interrogates 713,407 SNPs. The raw data files were analyzed with GenomeStudio2.0 software to retrieve the genotype dataset using the A/B allele Illumina notation. SNP coordinates used refer to the GRCh38 Genome Build. ***Quality control.*** Quality control (QC) of samples and SNPs was conducted in PLINK1.9 [20,21]. Samples were subjected to quality control, removing samples with a rate of missing genotype ≥ 2%, as well as samples that presented an excess heterozygosity as an index of contamination F < −0.05, calculated on the pruned dataset (pairwise correlations, window size 50 SNPs, sliding window of 5 positions, SNPs with r2 ≥ 0.2 removed). For SNP QC, SNPs with a rate of missing genotypes across samples≥ 1%, as well as SNPs diverging from Hardy-Weinberg equilibrium (*p*-value < 10^−6^), have been removed. In total, 57 samples and 665,191 variants passed quality control. ***Genotype covariates.*** PCA was performed in PLINK1.9 on the clean dataset (57 samples, 665,191 variants), pruned as described above, retaining the first 10 principal components (GenotypePC1-PC10) as covariates for downstream eQTL analysis to summarise population stratification. ***Ancestry Estimation.*** Ancestry estimation has been performed to infer the ethnicity of placental samples from Cochin and Anger cohorts. Briefly, the genotypes from the 1K Genomes Phase III release have been downloaded from The International Genome Sample Resource, mapped to the GRCh37 genome build, and have been used for imputation of allele frequencies across populations [22,23]. The set of SNPs in common between the two datasets has been established, and the two datasets merged using PLINK1.9. PCA was performed on the merged dataset, pruned as described above, including only autosomal chromosomes, for a total of 185,249 variants. Sample clustering along PC1 and PC2 have been used to extrapolate the unknown ethnicities of the placental samples based on their relative distance with the 1K Genomes Phase III samples. 

### 2.6. eQTL Analysis Workflow

The R package MatrixEQTL has been used to perform the eQTL analyses in R [24], by applying a linear multivariate model to test the contribution of each SNP genotype to gene expression levels. Two main approaches were pursued. The ALL COVARIATES dataset included the full set of 23 covariates (Batch, Group, Delivery Placental Sex, Gestational Age, Maternal Age, Parity, Ethnicity, the first 5 PCs from gene expression PCA, the first 10 PCs from genotype principal component analysis that summarises population stratification), allowing to identify the eSNPs correlating with changes of gene expression of the eGenes while minimizing spurious associations between SNPs and Genes [25]. The MINUS DISEASE dataset included all covariates but “Group” and Gene Expression PC5 given its strong correlation with the disease phenotype described by the “Group” variable. With this approach, the contribution of the disease variable on gene expression changes was not corrected for, allowing to investigate a set of genes for which the effects of disease on expression are important and would have otherwise excluded to the residual variance threshold, to identify SNPs that correlate with gene expression changes and could harbor interaction effects with the disease phenotype. 

For both approaches, the same workflow was followed. Each gene expression distribution was first normalized by a built-in function with rank (“average” method) followed by quantile normalization to reduce the impact of outliers. The normalized gene expression dataset was then modeled by multivariate linear regression, expressing gene expression for each gene in the dataset as a function of either the ALL COVARIATES or MINUS DISEASE set of covariates. The residual variance of the model can be thought of as an indication of potential genetic influence on gene expression levels [26,27]. Using incremental thresholds of residual variance from 0.5 to 0.95 with 0.05 increments, 10 gene sets termed “RV0.05” to “RV0.95” were defined and used as input gene expression datasets in eQTL analyses. 

### 2.7. Cis-QTL Analyses of Gene Expression Subsets

Gene subsets RV0.50 to RV0.95 and 417,114 SNPs (MAF ≥ 0.15) were analyzed for correlation with the LINEAR model parameter in MatrixEQTL [24]. We evaluated only the correlation of eGenes and eSNPs in *cis* (*cis*-eQTLs) given the limited power of our analysis and the small number of trans-QTLs previously identified in the placenta by a larger designed study [9]. To further reduce the total number of tests, the *cis*-distance has been set to 100 kb rather than the default 1 Mb [8]. The gene expression dataset has been normalized with MatrixEQTL prior to performing eQTL analysis, as described above. For the ALL COVARIATES Dataset analysis, 23 covariates (Batch, Group, Delivery, Maternal Age, Ethnicity, Gestational Age, Sex, Parity, first five PCs of gene expression PCA, first 10 PCs of genotype PCA) have been accounted for by the software when performing multivariate linear regression for each gene-SNP pair, while for the MINUS DISEASE dataset, 21 covariates have been included: Batch, Delivery, Maternal Age, Ethnicity, Gestational Age, Sex, Parity, first four PCs of gene expression PCA, first 10 PCs of genotype PCA. To correct for multiple testing, the in-built function in MatrixEQTL has been used that estimates FDR-adjusted *p*-values (*q*-values) for each gene-SNP pair with the Benjamini-Hochberg procedure. In this study, only SNPs with FDR ≤ 0.05 have been considered as statistically significant. The full lists of statistically significant eSNP-eGene pairs for both datasets are presented in Appendix A. 

### 2.8. Calculating Enrichment of Significant cis-QTLs for Each Subset

To choose the best *cis*-eQTL analytical design, defined by the gene expression subset used, the enrichment of significant *cis*-QTLs has been calculated. We define enrichment as the number of *cis*-eQTL per eGene. *Cis*-eQTL enrichment has been considered together with the total number of eGenes and eSNPs identified as a result of statistically significant *cis*-eQTLs (FDR ≤ 0.05) in order to find the optimal cut-off, where we see an increase of enrichment, without losing too many eGenes. For both ALL COVARIATES and MINUS DISEASE datasets, the input gene expression dataset RV0.85 was selected as optimal for the final cis-QTL analysis workflow and statistically significant cisQTLs were further characterized and are presented here. The list of input genes with relative individual expression levels as well as residual variances, are presented in Appendix A, respectively.

### 2.9. Calculating Overlap with Previous Studies

The statistical significance of the overlap with previous studies was calculated with Fisher’s exact test, with normal approximation, with a statistical significance threshold set at *p*-value ≤ 0.05 [28]. As reference total number of genes, we used N = 30,818; this number of placental expressed genes, with ≥0.1 RPKM (Read Per Kilobase of exon model per Million mapped reads), as defined in the latest high-quality study on placental transcriptome by Gong and coworkers [29]. The group carried out an RNA-sequencing analysis on 302 human placental samples, including messenger RNAs, long non-coding RNAs, as well as small and circular non-coding RNAs. This number is consistent with our study, where we observe 33,988 genes with mean fluorescent value (LOG_2_) ≥ 4.5. The total number of genes in the human genome, even though still lacking consensus, is reported to be higher than 35000, including coding and non-coding genes, as discussed in [30]. 

### 2.10. Calculating Statistical Significance of Interaction between Best-eSNP and Disease on eGene Gene Expression

To check whether an interaction exists between the genotype at the best-eSNP and the disease status on gene expression levels of the eGene we performed a linear regression in R and considered the linear model to be significant when the model *p*-value ≤ 0.05, similarly the effects of each variable on the model were considered significant when the coefficient *p*-value in the model was ≤ 0.05. The general formula of the linear regression model was: eGene ~ eSNP + Group + eSNP*Group. The linear regression was performed on the residuals of the gene expression dataset after multilinear regression with the MINUS DISEASE set of covariates to correct for confounding factors (the data had also initially been normalized by ranking and quantile normalization as described above).

## 3. Results

### 3.1. Transcriptome Identification of Confounding Variables

The transcriptome dataset is composed of 57 RNA samples that passed the quality controls from either the Cochin Hospital (Technologic Facility Gernom’IC), Angers University Hospital, or St George’s Hospital, University of London (Table 1). The transcriptome data were obtained using the Clariom D microarray (Affymetrix), which allows to analysis of the level of mRNAs at the exon level [12].

From the transcriptomic datasets, only genes with known GeneID and Description in the TAC (Transcriptome Analysis Console, Affymetrix) were kept, removing transcripts identified by the database AceView, as well as transcripts on the Y and mitochondrial chromosomes (n = 616 and 18, respectively), for a final total of 46,624 probes. Probes were further filtered by mean fluorescent value (LOG2) across all samples ≥ 4.5, eventually reducing the total number of probes to 33,988.

A new analysis strategy of our work was the removal of the covariates affecting gene expression; while some are interesting biological factors that could deserve interest, such as maternal age for instance, we decided to consider them all as confounders to identify solely eQTL that influence placental gene expression without obvious medical consequences. There were nine variables available for each sample that was included ab initio in the analysis (Batch, Cohort (St Georges, Cochin, or Angers), Disease Group, Mode of Delivery, Maternal Age, Ethnicity, Gestational Age at Delivery, Sex, Parity). Using PCA in R (PCATools) enabled us to define principal components (PCA axes). The 10 first PCA axes captured ~61% of the variability, while the two first axes, PC1 and PC2 only, concentrated ~34.5% of the variability. The correlation of the PC axes with the clinical variables is represented in Figure 2A. The weights of the clinical covariates could be estimated at 7.4%, 5%, 10.7%, 5.7%, 5.8%, 14.4%, 11.1%, 23.4%, and 23.5% for Parity, Sex, Gestational age, Ethnicity, Maternal Age, Delivery Mode, Disease Group, Cohort, and Batch, respectively. The total is above 100% since there are complex correlations between the different items; for instance, disease status is correlated with gestational age. Some axes were strongly correlated with the Group (i.e., the disease status), especially PC1 and PC5, the latter one being also intimately correlated with the gestational age.

In addition, The R package SVA was used to identify possible surrogate variables (additional unknown sources of variation in the dataset). We collated all variables in a correlation matrix that was calculated in R, using as input the nine clinical and experimental variables, the first 10 PCs of gene expression PCA, and four identified surrogate variables to identify the final set of variables correlated with global transcriptomic changes. A correlation coefficient cut-off threshold of |0.9| was used to remove colinear variables [19]; see Figure 2B. For the final choice of covariates included in the eQTL analysis, we kept the five first PCs, “Batch”, “Group”, “Delivery”, “Maternal Age”, “Ethnicity”, “Gestational Age” and “Sex”. The covariate “Cohort” was removed since it was colinear with “Batch”.

### 3.2. Genotyping and Population Stratification

The final genotype dataset included 665,191 variants for the 57 samples, of which we kept SNPs with a Minor Allele Frequency > 0.15 (417,114 were kept). The legislation in France forbids the collection of ethnic data. Nevertheless, with the genotype information, we were able to classify our samples using PCA alone and with reference to the 1000 genome project (Appendix A). Seven placental samples were clearly of African origin, 42 were of European origin, and eight were mixtures between African and European backgrounds. The first 10 PCs from the genotype principal component analysis were included as covariates in the eQTL analysis to account for variations due only to population stratification. 

### 3.3. Optimal Feature Selection for the eQTL Analysis

We first explored the effect of varying the number of input genes for the eQTL analysis to identify the optimal gene sets in order to achieve a satisfactory balance between the number of statistically significant cis-eQTLs (FDR ≤ 0.05), the number of identified eGenes and cis-eQTL enrichment (in 100 kb proximal to the gene), defined as the number of eSNPs/eGene. For this, we used the concept of residual variance (RV) to reduce the number of tested genes. 

Given our limited sample size, it was our primary goal to be able to define the set of genes to be used as input in the eQTL analysis that would give us the optimal experimental design to identify significant *cis*-eQTLs, reducing as much as possible the number of tests, without missing out potentially relevant genes. The eQTL workflow that we used was based on multivariate linear modeling, which expresses the variance observed in the expression levels for each gene in the function of the input covariates. Therefore, the RV is a measure of potential genetic influence on gene expression levels, amongst other variables [26,27]. The summary of the threshold optimization for analysis is presented is Appendix A. While the number of significant eGenes decreased from 153 to 22, when the RV raised from 0.5 to 0.95, the number of significant SNP per gene increased from 4.00 to 8.91.

We selected an RV threshold of 0.85 for further analysis as a good compromise between having significant but relevant eQTL, albeit other thresholds could certainly be chosen. At this threshold, at least six SNP per gene were significant, meaning that the association with gene expression is not merely due to linkage disequilibrium. The analysis at this threshold rested upon 3201 genes, 417114 SNPs, a multivariate model with 23 covariates, and a window of ±100 kb around each gene, following the experimental choice of Kikas et al. [8], which justified this window by the fact that relevant regulatory SNPs are mostly in the vicinity (~100 kb) of the gene. These parameters resulted in the discovery of 279 statistically significant placental SNPs (*p* < 1.24 × 10^−4^, FDR < 0.05) proximal to 43 eGenes (Figure 3, Table 2). The full list of statistically significant eGene-eSNP pairs is provided in Appendix A. 

In Table 2, the novel genes discovered are presented in yellow, while the genes in bold correspond to the most stringent threshold and to a Bernouilli genome-wide correction which does not consider the possible LD relating to successive SNPs. When this is considered, the Bernouilli correction corresponds to a limited number of independent tests, leading to a threshold of ~10^−4^ instead of 10^−8^ (blue and red line, respectively, in Figure 3A). We report 22 novel placental eQTL, while 51% of the placental eGenes identified here have been previously described as placental cis-eQTLs in the three placental studies available to date (n = seventeen for the Peng et al. dataset, n = eight for Delahaye et al., and n = five for Kikas et al.) [7,8,9]. Fisher’s exact tests of the overlaps between datasets were all statistically significant with *p*-values < 0.05, respectively *p* < 5.83 × 10^−7^, *p* < 1.88 × 10^−6^, and *p* < 2.75 × 10^−8^, validating our experimental approach. 

Interestingly, out of the 279 cis-QTL found, 37 corresponded to SNPs located between two genes modified at the expression level. On chromosome 6, 24 eSNPs were found to be shared between TOB2P1 and ZSCAN9 out of their 41 and 42 cis-eQTLs, respectively (Figure 3B). On chromosome 11, two different regions of interest were identified, between LOC646029 and AQP11, that shared 10 eSNPs, out of 11 and 12 cis eQTLs, respectively. Finally, three eSNPs associated with CARD17 expression deregulations correlated also with upstream transcript CASP1P2, which had a total of 11 cis-eQTLs (Figure 3B). These observations strongly suggest that the SNPs influence gene expression in the chromosome region, either by modifying an enhancer acting on two genes close by or by being transcribed in a non-coding RNA having a cis-regulatory effect on transcription.

### 3.4. Identifying eQTLs Involved in the Disease by a Subtraction Strategy

The same cis-eQTL analysis was performed de novo, suppressing the parameter ‘Disease’ from the confounding factors, as well as PC5 (Figure 2), which were very highly correlated. This allowed us to include in the analysis genes for which the disease has a strong influence on changes in gene expression levels, reducing the overall residual variance after modeling (the number of genes increased from 3201 to 3279, Figure 4). However, these genes are also believed to be the most interesting in terms of understanding pregnancy pathologies. For this reason, we aimed to investigate the contribution of genotype changes on these genes in the context of disease to be able to also detect potential genotype-disease interactions that would suggest the presence of conditional eQTLs [31] Interestingly, in this case, the number of identified eGenes was always higher than when the ‘Disease’ was used as a covariate. At the chosen threshold of 0.85 for the residual variance (Figure 4), the number of eGenes identified increases from 43 to 54, with the number of statistically significant (FDR ≤0.05) eSNP increasing from 279 to 307, as represented in Figure 4. The interpretation of this figure is that 16 SNPs are specific only to the disease, 38 are influenced by the covariates and the disease, while amongst the five specific to the dataset analyzed with all the covariates, four did not pass the RV threshold, MLLT10, HJURP, MIR4527, and Y_RNA. The last one, FGF19, is present in both datasets, but when the disease is not considered as a covariate, FGF19 is no longer observed as an eGene. The full list of statistically significant eGene-eSNP pairs is presented in Appendix A, while the top-eSNP is listed in Appendix A. 

The 16 eGenes that were specific to the disease are presented in Figure 5 and Table 3. Several analyses of gene ontology were carried out using these 16 eGenes (disease-specific) or the 54 eGenes (placental-specific), including the 38 genes that were also presented in the first analysis using String (https://string-db.org/, accessed on 1 July 2021). All these analyses failed to identify a significant grouping of genes, suggesting that the couples eSNP-eGene that we could identify correspond to various aspects of placental biology and physiopathology and not to several specific categories or gene ontologies. Not surprisingly, however, some of these genes have been previously identified as deregulated in pregnancy pathologies (Table 3).

While previously identified eQTLs, such as influencing ZSCAN9 and ERAP2, have shown a strong genetic influence on their regulation, the implication of this on placental function remains elusive. In Figure 6, Figure 7 and Figure 8, we show the levels of expression stratified according to genotype of the most significant eSNP for a subset of eGenes that presented a pronounced conditional response depending on disease group (Control vs Disease) and have been previously linked to placental function and diseases such as IL36R, FUT10, PTTG1, CBLB, and DNAJC15. Here, we plot the raw data for the expression levels; however, during analysis, the data is first normalized and then corrected for the covariates. Table 4 summarizes the results of Genotype*Disease interaction testing for each of the eGenes identified and its most significant eSNP. Interestingly, ZSCAN9, AQP11, LOC646029, FUT10, NDUFS5, and C8ORF89 showed statistically significant (*p*-value ≤ 0.05) interaction between the effects of disease status and the eSNP genotype on the expression of the eGene. This suggests the existence of a complex interaction between the gene and the phenotype of the disease. 

## 4. Discussion

In this paper, we attempted a novel approach to uncover relevant associations between genetic variants and gene expression in the placenta from healthy and pathological pregnancies. We started by carefully identifying covariates that may have masked important results in previous studies. As observed previously, gestational age is a major covariate and is strongly associated with disease status. In PCA analysis, the fifth axis is the most strongly associated with gestational age (r = 0.49, Figure 2A), and the most strongly associated to placental disease (r = −0.47, Figure 2A). The inverse correlation between the two features is consistently seen in many transcriptomic studies comparing normal and pathological placentas. The problem is that, almost by definition, placentas from pregnancy complications diseases are usually obtained earlier in gestation than from normal pregnancies. The originality of our approach is as follows: we performed two separate searches for eQTLs, one using all covariates available (including disease status) and one without the disease status used as a covariate, assuming that the difference in the discovery will be due more or less exclusively to the disease. At the initiation of this work, we performed a thorough characterization of the covariates influencing gene expression, and while we could not fully distinguish between “disease group” and “gestational age” given their strong correlation with PC5, we could partially account for gestational age confounding by statistically correcting for PC2 which is strongly correlated (0.39, *p* val < 0.01), but not with “disease group”.

In the first part of our results which included “disease group” as a covariate, we expanded the current knowledge of genetic modulation of gene expression in the placenta, adding to the relatively limited number of previous studies in the field [7,8,9,12]. We identified 279 placental *cis*-eQTLs (FDR ≤ 0.05) in 57 human placental samples, correlating with expression changes of 43 unique eGenes, among which 22 constituted novel eGenes over the previous studies. One hundred and eleven (111) of the total 279 *cis*-eQTLs, including the most significant eSNPs, were associated with previously described placental eGenes *ZSCAN9*, *PSG7*, *TOB2P1*, *ERAP2*, and *AQP11*, corroborating the validity of our analysis. Interestingly, for the first time in the placenta, we describe *cis*-eQTL eSNPs that correlate with neighboring transcripts, identifying potential placental regulatory hotspots. On chromosome 11, a novel eGene *LOC646029* and downstream *AQP11* shared 10 eSNPs, while the three eSNPs associated with *CARD17* expression levels were also eSNPs for novel placental eGene *CASP1P2*. Similarly, 24 out of the 42 eSNPs associated with *ZSCAN9* were found to correlate with changes in expression of the upstream transcript *TOB2P1* on chromosome 6. *CASP1P2* and *TOB2P1* are both pseudogenes, which implies that they may not encompass functional open reading frames and that changes in their expression could not have an impact on placental function as proteins. However, the changes in their expression levels as well as in neighboring genes as a result of genetic changes, highlight these regions as potential placental regulatory sites. As additional evidence of the relevance of these regions to placental function, genotype changes in these loci show a statistically significant interaction with the disease status, resulting in significant changes in the expression of ZSCAN9 and LOC646029, as well as AQP11. 

Kikas and coworkers extensively reviewed previous studies on genetic regulation of gene expression in the placenta, including gene-candidate approaches, as well as the three genome-wide studies mentioned above [40]. Peng and coworkers combined RNA-sequencing and genotype data of 159 healthy placentas, laying the foundations for eQTL analyses in the placenta [9]. They report only the top eSNP of each eGene, therefore identifying 3218 *cis*-eQTLs correlating with 3218 unique transcripts and 35 *trans*-eQTLs (*cis*-distance between eSNP and eGene < 500kb) with an FDR threshold, defined by permutations, of 0.1%. This limited number of *trans*-QTLs, even with a sample size that is more than double one of other studies in placental eQTLs, prompted to restrict further efforts to *cis*-QTL identification alone. Delahaye and coworkers, expanded the repertoire of placental *cis*-eQTL by identifying 985 eSNPs, corresponding to 615 eGenes in 80 healthy placentas, as well as investigating methylation-QTLs in 300 placenta samples [7]. Finally, Kikas and coworkers, with a total of 40 samples, identified 199 significant *cis*-eQTLs (FDR < 0.05) corresponding to 63 unique genes in a window of 100 kb around each gene [8]. Both studies showed considerable overlap with [9], respectively, for 62% and 80% of the eGenes. However, if we consider the percentage of unique eGenes, not replicated across all studies, 88%, 37%, and 21% of eGenes were identified, respectively, by Peng et al. [9], Delahaye et al. [7] and Kikas et al. [8].

When comparing our results with previous studies, the overall level of overlap between eGene lists is 51%, which could be due to differences in population stratification, sample size as well as differences in analysis workflow. In particular, the number of overlapping eGenes is n = seventeen for the [9] dataset, n = eight for [7], and n = five for [8]. The most significant overlap is with the work of [8]; Fisher’s exact tests *p* < 2.75 × 10^−8^, 1–2 orders of magnitude more significant than the overlap with the two other studies. This could be explained by the fact that in terms of analytical design, our study is closer to the one of [8]. In both analyses, cis-QTLs were analyzed in a window of 100 kb, for a total number of variants of 417,114, in the same order of magnitude, similarly; the sample sizes are also close with 40 and 57 placentas in [8] and our study respectively. Probably the most crucial similarities are the use of the same software for the eQTL analysis (i.e., MatrixEQTL in R [24]) and the inclusion of placentas from pregnancies complicated by PE. The use of the same analytical software improves the replicability of the results since the statistical tests are the same and can have a huge impact on which eSNP-eGene pairs pass the threshold of significance. 

Similarly, the presence of PE samples implies a similar pattern of gene expression; however, in our study design, we did consider “disease group” as a confounding factor and included it as a covariate, which should reduce the effects on the output.

With our analysis comparing the two datasets, the ALL COVARIATES vs. MINUS DISEASE, that differ only by removing the disease as a confounding covariate in the second, we propose a new strategy for finding eQTL that are relevant in pathological pregnancies. In particular, this approach allowed us first to validate our study design as well as sample quality by being able to compare our findings with the existing literature. In the second instance, we were able to further investigate the role of genotype changes on the expression of genes for which disease has a relevant influence and could therefore be important for placental function. 

Genome-Wide Association Studies (GWAS) use large sample sizes (in the tens of hundreds) in an attempt to identify the enrichment of variant alleles in the disease group compared to healthy controls. GWAS conducted in the placenta has identified SNPs upstream of the FLT1 locus, rs4769613, as harboring the risk allele (T) associated with PE [41]. Despite the limited functional understanding offered by GWAS approaches, the use of combinatorial approaches that exploit knowledge from different types of datasets helps draw these functional conclusions [42,43,44]. Recently, the rs4769613 locus was genotyped in a cohort of placental samples, including 57 PE and 277 control; the presence of the risk allele correlated with changes in the placental gene expression level of FLT1 only in the PE group, defining rs4769613 as a conditional eQTL [31]. This suggests that the risk variant, when present in the placentas of mothers with PE, favors the increase in expression of the FLT1 gene and, therefore, could be located in a regulatory region for the gene itself, targeted by regulatory factors that are expressed specifically in the PE placenta.

In our work, we identified multiple genes that presented a promising ‘genotype and disease’ interaction, which could therefore represent placental conditional eQTLs relevant to disease etiology. The genes found during this stage are supposed to be influenced by the disease either exclusively or mainly but with an influence of other factors.

Among these, AQP11, IL36R, FUT10, PTTG1, CBLB, and DNAJC15 are interesting both in terms of behavior, as well as the documented role in placental function.

Among the ‘disease exclusive’ genes represented in detail (in green boxes), there are PTTG1, DNAJC15 (Figure 6), TAS2R64P and LINC00654 (Figure 7), C8ORF89 and ERICH1 (Figure 8). The profiles of expression are very different; for instance, in the case of ERICH1, the SNP influence of the profile is the same way in disease and controls, but overall in the disease, the expression is systematically inferior. By contrast, C8ORF89 has an SNP-dependent profile in controls but not in disease; PTTG1 is overexpressed in disease but only rather in specific SNP configuration.

PTTG1 has been involved in trophoblast invasion through the regulation of matrix metalloproteases MMP2 and MMP9 by regulating integrin/rho signalling [34]. AQP11 belongs to the family of water transporters through the cellular and endoplasmic reticulum membrane. It has recently been implicated in redox regulation, and other members of the same family, such as AQP, were found to be involved in the mechanism of aspirin action in preventing PE [45,46,47]. IL36RN is highly expressed in the placenta, where it could be involved in trophoblast proliferation [48]. FUT10, involved in the fucosylation of proteins, has been shown to promote the binding of TNF to its receptor, thus promoting the inflammatory reaction, which is relevant for placental diseases [49]. Interestingly, we identified a significant interaction between disease and genotype on the levels of FUT10. DNAJC15 is reported as an important factor in mitochondrial function and regulation of the respiratory chain. It has been abundantly demonstrated by us and many others how these cascades of oxidative phosphorylation and oxidative stress are pivotal in placental diseases [50,51]. CBLB has been recently shown to influence Natural Killer cell differentiation [52] and Uterine NK cells are pivotal for efficient implantation. In summary, all these genes may be involved in placental disease by a contribution to various pathways, which explains why they are not explicitly regrouped in a biological function.

## 5. Study Limitations and Conclusions

Amongst the limitations of our study, there is, of course, the sample size, which makes our data a preliminary incitement that needs confirmation in further studies. The choice to subtract extensively the covariate effect encompasses a risk of missing positive hits (Type II error), but our objective was to decrease the Type I error and minimize the finding of false positives. Some additional analyses could be performed by removing some of the covariates to detect eSNPs influenced by sex, but this is beyond the objective of our current work. Furthermore, among the control placentas, one-third were from natural deliveries, which we included in our study by adding the delivery mode as an additional covariate). Gestational age was found to induce 10% of the variability, consistent with previous observations in the field. Subtracting its effect and using it as a covariate might be considered a mathematical trick that could mask complex interactions between the disease effects and the gestational age effects. Nevertheless, it is currently difficult to adopt a better strategy in the field of placental disease.

The approach based on two-hits, using unusually the disease as a covariate and then adding it afterward, allows by subtraction to identify novel couples of SNP-eGenes that may be of interest for placental disease studies or other cases of human diseases where confounding factors are important. We hope that our work will be a starting point for researchers using other complex datasets available for analysis.

## Figures and Tables

**Figure 1 cells-12-00578-f001:**
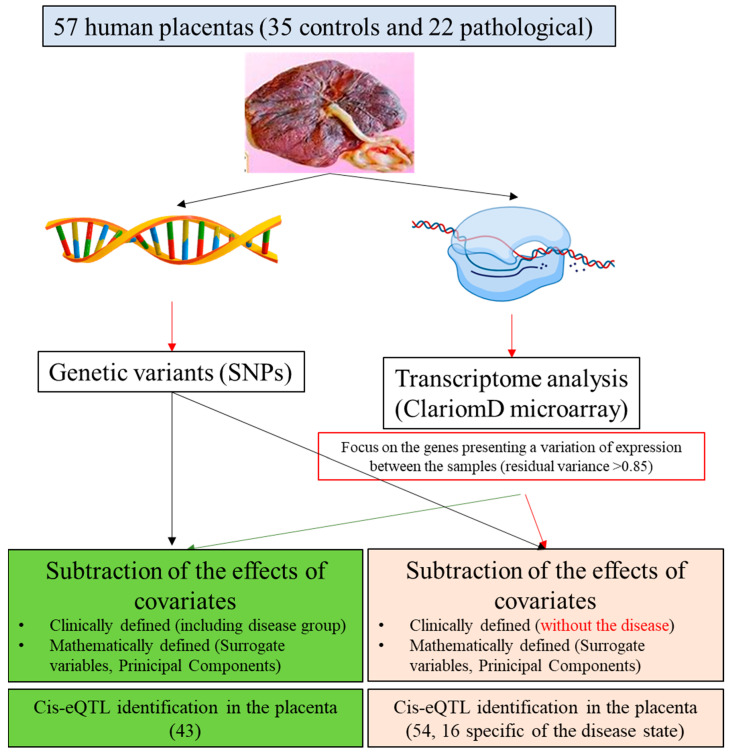
Overview of the experimental design. SNP = Single Nucleotide Polymorphism; PCA = Principal Component Analysis; SVA = Surrogate Variable Analysis; eQTL = expression Quantitative Trait Locus.

**Figure 2 cells-12-00578-f002:**
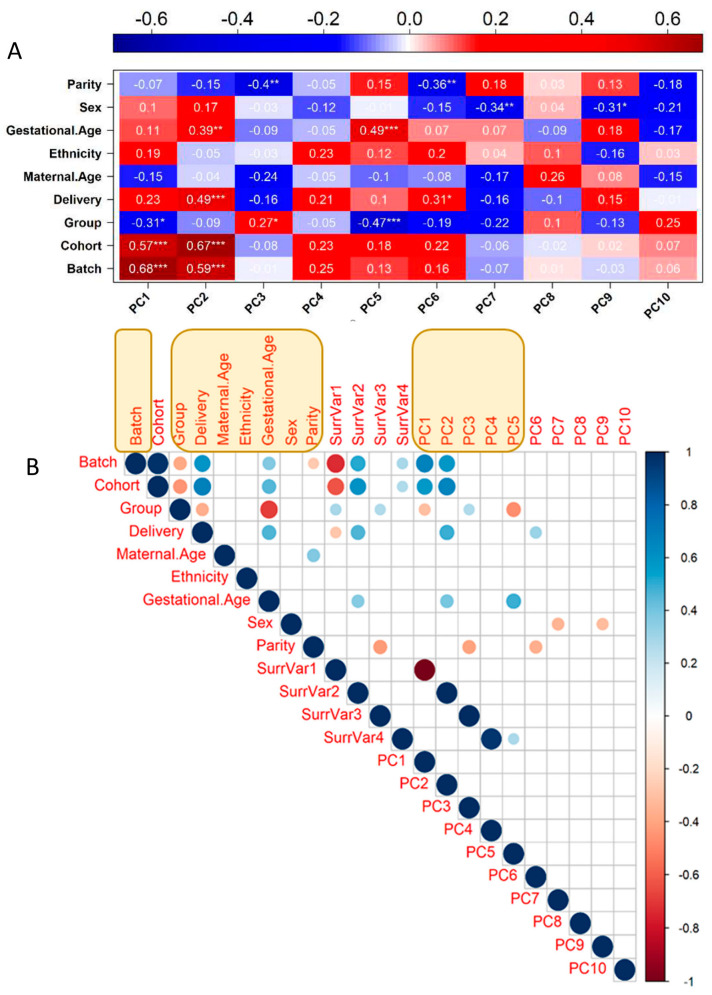
Definition of covariates and surrogate variables potentially affecting the gene expression dataset. (**A**) Eigencorplot expresses the results of Pearson correlation between the clinical and technical variables and the first 10 principal components obtained from the PCA performed on the gene expression dataset. A negative correlation is expressed in blue, and a positive correlation is expressed in red. Within each pair the correlation coefficient is displayed with statistical significance. (* = *p*-value ≤ 0.05, ** = *p*-value ≤ 0.01, *** = *p*-value ≤ 0.001). (**B**) Correlation matrix between the 9 clinical and experimental variables, first 10 PCs of gene expression PCA, and 4 identified surrogate variables, having effects on global transcriptomic changes. Only statistically significant correlations (*p*-value ≤ 0.05) are displayed as dots of increasing size as a measure of the correlation coefficient. Positive correlations are displayed in blue, and negative correlations are displayed in red. In yellow boxes are the variables that were kept as covariables for subtracting their effect (see text).

**Figure 3 cells-12-00578-f003:**
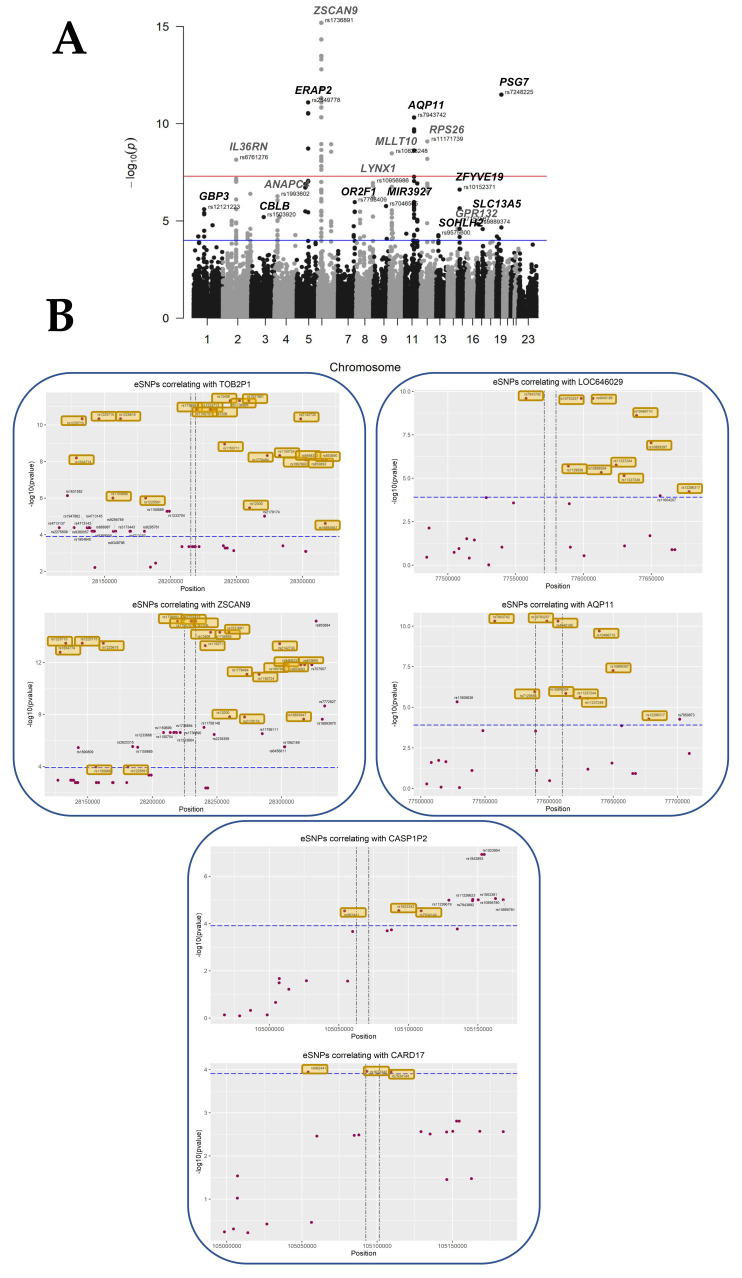
Overview of the cisQTL results for the ALL COVARIATES dataset. (**A**) Manhattan plot showing each tested SNP as a dot in its genomic position on the x-axis, stratified by chromosomes. The y-axis expresses the -LOG10 of the *p*-value of the interaction with the eGene. Suggestive line in red shows the 10 × 10^−8^ threshold, while in blue, the threshold corresponding to a *p*-value of 0.05 adjusted for the number of tests (False Discovery Rate) = 1.24 × 10^−4^. (**B**) Plots showing the localization of the tested SNPs within the genomic regions on chromosome 6, containing ZSCAN9 and TOB2P1, and chromosome 11, where AQP11, LOC646029, CASP1P2, and CARD17 are located. The x-axis expresses the genomic coordinates, while in the y-axis, the -LOG10 of the *p*-value, the horizontal line sets the level for FDR ≤ 0.05. The vertical lines represent the start and end sites for each of the genes. eSNPs that were found to be in common between eGenes in the same region are highlighted in yellow.

**Figure 4 cells-12-00578-f004:**
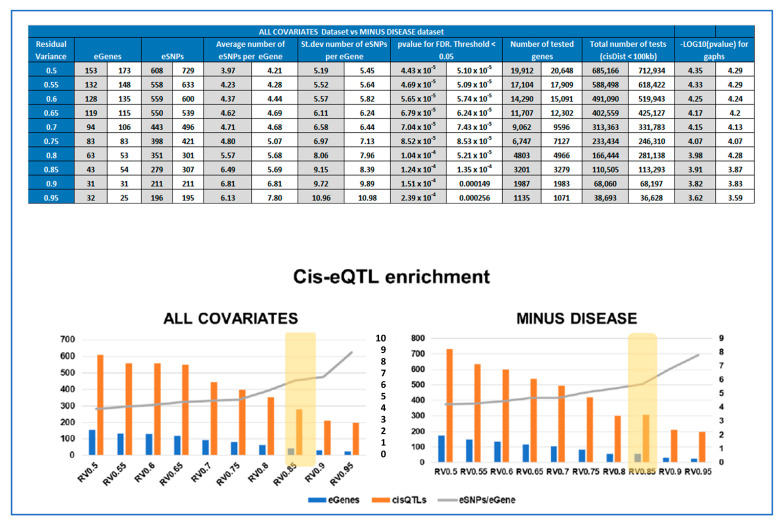
Comparing the experimental parameters for the ALL COVARIATES and MINUS DISEASE eQTL datasets. (**TOP**) Table comparing the parameters of cis-eQTL analyses performed for different Residual Variance (RV) thresholds (0.5-0.95) for ALL COVARIATES and MINUS DISEASE datasets. Depending on the RV threshold, different lists of genes were included in the cisQTL analysis, resulting in 10 output cis-QTL datasets for each approach, ALL COVARIATES in grey and MINUS DISEASE in white. (**BOTTOM**) Cis-eQTL enrichment graphs for each approach at the different thresholds of residual variance tested. Showing the number of significant eGenes (blue), eSNP (orange), and enrichment expressed as number of eSNP/eGene in grey.

**Figure 5 cells-12-00578-f005:**
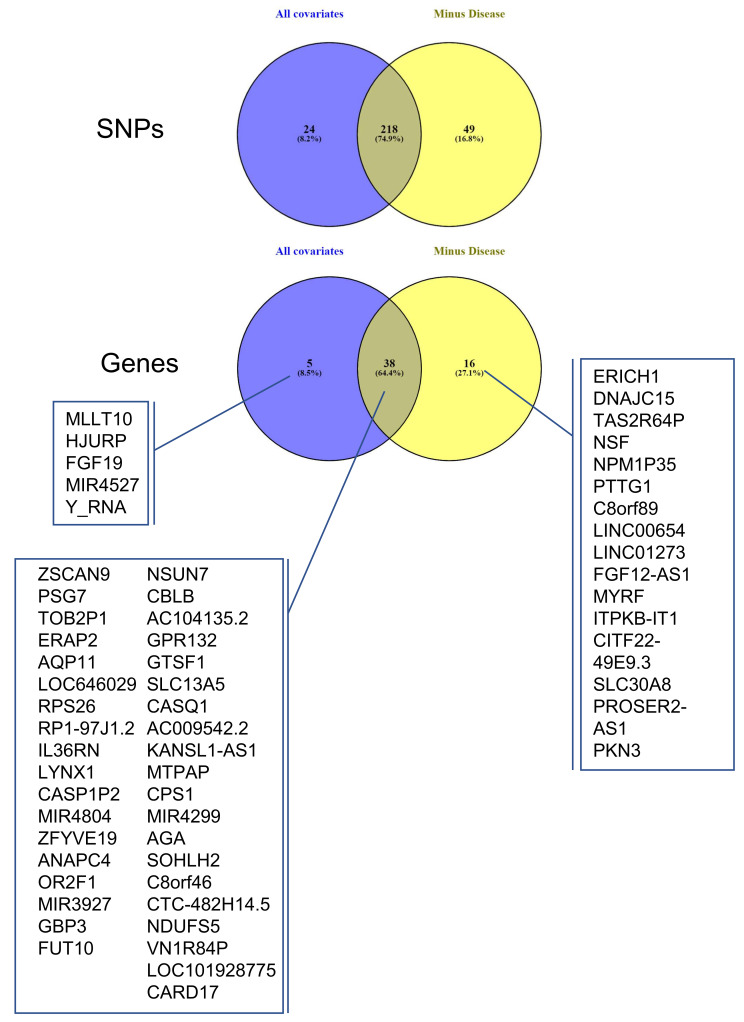
Overlap between cis-QTL datasets ALL COVARIATES and MINUS DISEASE. Venn diagrams showing the overlap between the eSNPs and eGenes identified as statistically significant (FDR ≤ 0.05) in ALL COVARIATES (blue) and MINUS DISEASE (yellow).

**Figure 6 cells-12-00578-f006:**
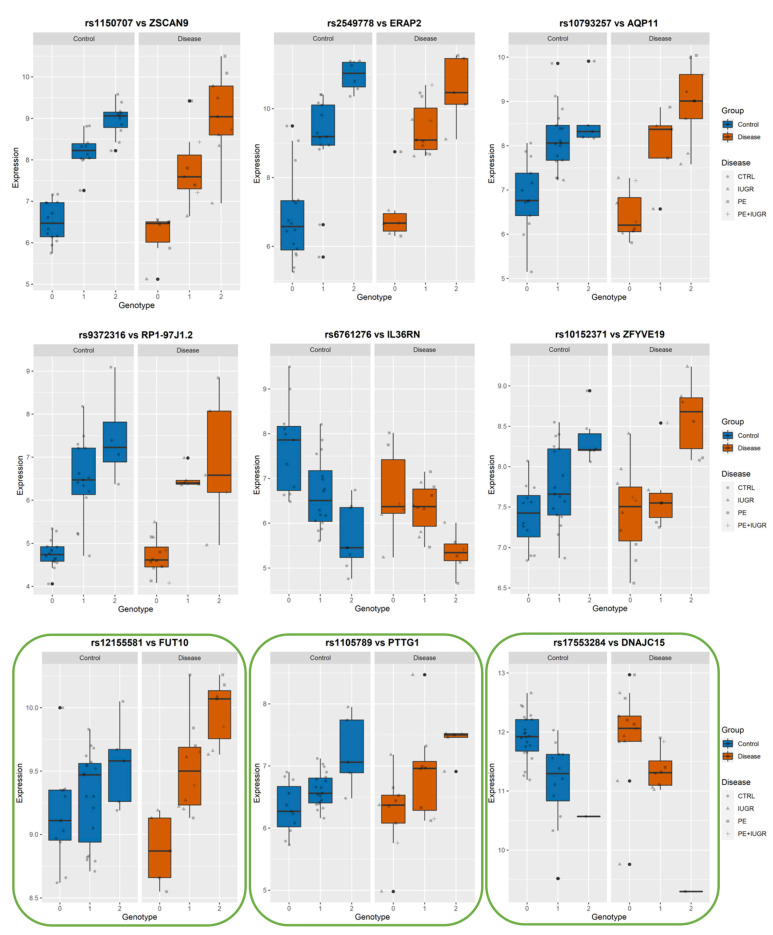
Boxplots of eGene expression in relation to the best-eSNP genotype, stratified by Group: Control vs. Disease. eGenes ZSCAN9, ERAP2, AQP11, RP1-97J1.2, IL36RN, ZFYVE19, FUT10, PTTG1, DNAJC15.The genotype of each SNP is expressed as 0, 1, 2, and the gene expression data corresponds to the raw data as mean fluorescence value. CTRL = Control; IUGR = Intra-uterine Growth Restriction; PE = Preeclampsia. Genes discussed in the text, because of their relevance to placental biology, are highlighted in green.

**Figure 7 cells-12-00578-f007:**
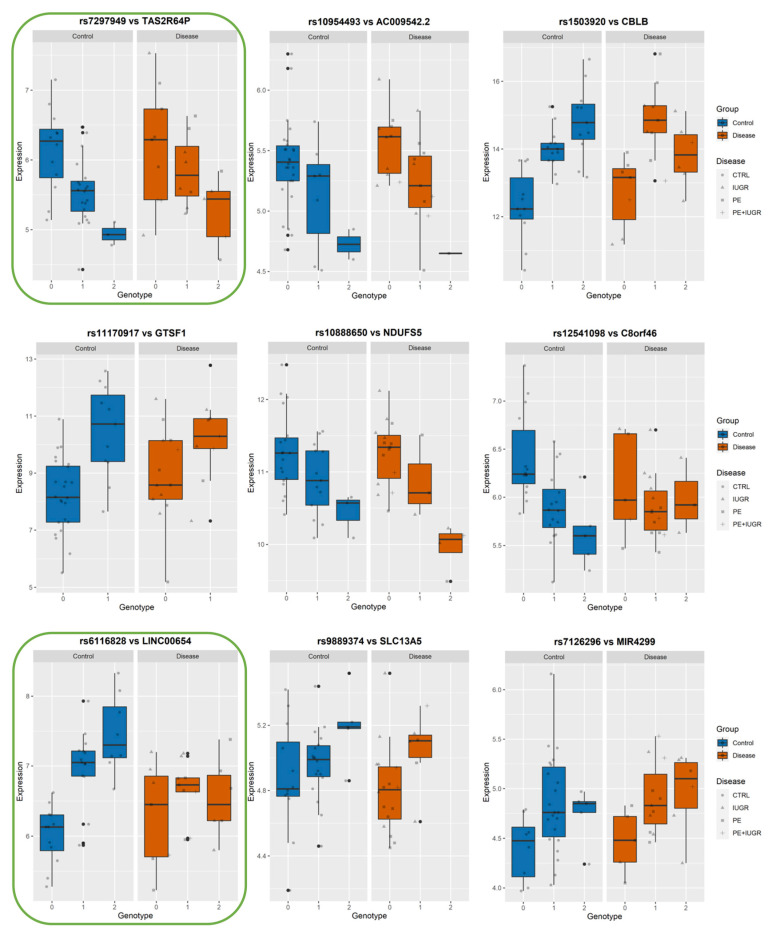
Boxplots of eGene expression in relation to the best-eSNP genotype, stratified by Group: Control vs. Disease. eGenes TAS2R64P, AC009542.2, CBLB, GTSF1, NDUFS5, C8orf46, LINC00654, SLC13A5, MIR4299.The genotype of each SNP is expressed as 0, 1, 2, and the gene expression data corresponds to the raw data as mean fluorescence value. CTRL = Control; IUGR = Intra-uterine Growth Restriction; PE = Preeclampsia. Genes discussed in the text, because of their relevance to placental biology, are highlighted in green.

**Figure 8 cells-12-00578-f008:**
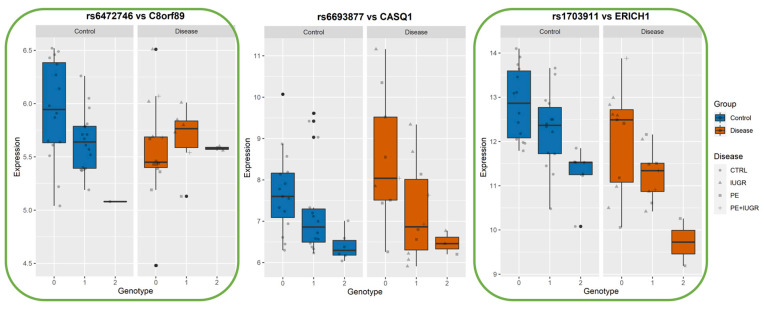
Boxplots of eGene expression in relation to the best-eSNP genotype, stratified by Group: Control vs Disease. eGenes C8orf89, CASQ1, ERICH1. The genotype of each SNP is expressed as 0, 1, 2, and the gene expression data corresponds to the raw data as mean fluorescence value. CTRL = Control; IUGR = Intra-uterine Growth Restriction; PE = Preeclampsia. Genes discussed in the text, because of their relevance to placental biology, are highlighted in green.

**Table 1 cells-12-00578-t001:** **Summary statistics of human placental samples sorted by disease**.

Disease Group	CONTROLS	PE	PE + IUGR	IUGR
Delivery mode (Caesarian/Natural)	23/12 (61.4%)	9/0 (15.8%)	3/0 (5.3%)	10/0 (17.5%)
Maternal age (years)	34.0 ± 3.9	34.2 ± 6.0	35.3 ± 2.4	32.1 ± 6.6
Ethnicity (Afr/Eur)	8/27	5/4	1/2	1/9
Gestational age (years)	39.2 ± 1.2	34.9 ± 2.6	30.0 ± 2.5	31.0 ± 2.8
Sex (M/F)	17/18	4/5	2/1	3/7
Parity	1.9 ± 1.5	1.2 ± 1.1	2.0 ± 1.4	1.2 ± 0.4

**Table 2 cells-12-00578-t002:** **List of eGenes, with the most significant cis-eQTL for the ALL COVARIATES dataset.** The table lists the total number of eGenes correlating with the identified significant cis-eQTLs; for each eGene the total number of eSNPs are listed, and the details of the Best-eSNP reported. The Best-eSNP, in this case, is defined as the most significant cis-eQTL associated with each eGene (smallest recorded *q*-value). Beta corresponds to the slope coefficient for the multivariate linear model and is an estimation of the eSNP effect size on gene expression variance. Novel placental eGenes are highlighted in yellow. If the eGene had been previously identified as such, the studies are specified in the “Previous study” column.

Gene Symbol	Description	eSNPs	Best-eSNP	*p*-Value	*q*-Value	Beta	Chr	Pos.	Previous Study
ZSCAN9	zinc finger and SCAN domain containing 9	42	rs1150707	6.46 × 10^−16^	1.43 × 10^−11^	18.80	6	28229827	[8]
PSG7	pregnancy specific beta-1-glycoprotein 7 (gene/pseudogene)	3	rs7248225	3.20 × 10^−12^	1.68 × 10^−8^	27.40	19	42918847	[7,8,9]
TOB2P1	transducer of ERBB2, 2 pseudogene 1	41	rs13408	4.79 × 10^−12^	2.21 × 10^−8^	−17.74	6	28244970	[9]
ERAP2	endoplasmic reticulum aminopeptidase 2	13	rs2549778	8.01 × 10^−12^	3.28 × 10^−8^	19.34	5	96868551	[7,8,9]
AQP11	aquaporin 11	12	rs10793257	4.80 × 10^−11^	1.29 × 10^−7^	20.84	11	77598488	[7,8,9]
LOC646029	uncharacterized LOC646029	11	rs10793257	2.48 × 10^−10^	6.09 × 10^−7^	20.61	11	77598488	
RPS26	ribosomal protein S26	9	rs11171739	8.12 × 10^−10^	1.95 × 10^−6^	18.98	12	56076841	[7]
RP1-97J1.2	putative novel transcript	7	rs9372316	1.14 × 10^−9^	2.62 × 10^−6^	21.02	6	112000000	
MLLT10	myeloid/lymphoid or mixed-lineage leukemia	15	rs10828248	3.35 × 10^−9^	6.73 × 10^−6^	−16.45	10	21535690	
IL36RN	interleukin 36 receptor antagonist	25	rs6761276	7.09 × 10^−9^	1.31 × 10^−5^	−17.30	2	113000000	[8,9]
LYNX1	Ly6/neurotoxin 1	9	rs10956986	1.13 × 10^−7^	1.47 × 10^−4^	16.92	8	143000000	[9]
CASP1P2	caspase 1 pseudogene 2	11	rs1023954	1.18 × 10^−7^	1.50 × 10^−4^	19.23	11	105000000	
MIR4804	microRNA 4804	5	rs2253215	1.26 × 10^−7^	1.56 × 10^−4^	23.89	5	72952041	
ZFYVE19	zinc finger, FYVE domain containing 19	6	rs10152371	2.43 × 10^−7^	2.63 × 10^−4^	16.13	15	40811095	[7,9]
HJURP	Holliday junction recognition protein	4	rs2361506	4.05 × 10^−7^	4.11 × 10^−4^	18.92	2	234000000	
ANAPC4	anaphase promoting complex subunit 4	10	rs1993602	5.38 × 10^−7^	5.41 × 10^−7^	−15.90	4	25413484	[9]
OR2F1	olfactory receptor, family 2, subfamily F, member 1 (gene/pseudogene)	5	rs7798409	1.08 × 10^−6^	1.00 × 10^−3^	−20.39	7	144000000	
MIR3927	microRNA 3927	1	rs7046565	1.72 × 10^−6^	1.56 × 10^−3^	16.92	9	110000000	
GBP3	guanylate binding protein 3	5	rs12121223	2.50 × 10^−6^	2.14 × 10^−3^	−18.83	1	89015900	[9]
FUT10	fucosyltransferase 10 (alpha (1,3) fucosyltransferase)	5	rs7018447	3.31 × 10^−6^	2.67 × 10^−3^	16.61	8	33467537	[9]
NSUN7	NOP2/Sun domain family, member 7	3	rs2437317	6.06 × 10^−6^	4.29 × 10^−3^	25.15	4	40789990	
CBLB	Cbl proto-oncogene B, E3 ubiquitin protein ligase	1	rs1503920	6.32 × 10^−6^	4.45 × 10^−3^	15.58	3	106000000	[8,9]
AC104135.2	novel transcript	3	rs7573356	9.72 × 10^−6^	6.43 × 10^−3^	−20.13	2	74941537	
GPR132	G protein-coupled receptor 132	2	rs7157567	1.05 × 10^−5^	6.70 × 10^−3^	18.27	14	105000000	
GTSF1	gametocyte specific factor 1	1	rs11170917	1.31 × 10^−5^	7.98 × 10^−3^	24.42	12	54472204	[9]
SLC13A5	solute carrier family 13 (sodium-dependent citrate transporter), member 5	1	rs9889374	1.46 × 10^−5^	8.69 × 10^−3^	17.28	17	6662428	
CASQ1	calsequestrin 1 (fast-twitch, skeletal muscle)	1	rs6693877	1.67 × 10^−5^	9.80 × 10^−3^	−14.62	1	160000000	[9]
AC009542.2	novel transcript, antisense to WDR91	3	rs10954493	1.87 × 10^−5^	1.08 × 10^−2^	−18.52	7	135000000	
KANSL1-AS1	KANSL1 antisense RNA 1	1	rs17585974	2.62 × 10^−5^	1.46 × 10^−2^	24.81	17	46171833	
MTPAP	mitochondrial poly(A) polymerase	2	rs1762598	4.17× 10^−5^	2.15 × 10^−2^	−15.63	10	30352501	
CPS1	carbamoyl-phosphate synthase 1	3	rs918233	4.27 × 10^−5^	2.20 × 10^−2^	−16.64	2	211000000	[9]
MIR4299	microRNA 4299	1	rs7126296	4.53 × 10^−5^	2.32 × 10^−2^	17.44	11	11556715	
AGA	aspartylglucosaminidase	2	rs4690523	5.31 × 10^−5^	2.63 × 10^−2^	−15.80	4	177000000	[9]
SOHLH2	spermatogenesis and oogenesis specific basic helix-loop-helix 2	4	rs9575600	5.41 × 10^−5^	2.64 × 10^−2^	16.53	13	36223006	[7]
C8orf46	chromosome 8 open reading frame 46	1	rs12541098	6.17 × 10^−5^	2.90 × 10^−2^	−15.38	8	66424147	[7,8,9]
CTC-482H14.5	novel transcript, antisense to PTPRS	1	rs2251127	6.29 × 10^−5^	2.90 × 10^−2^	16.34	19	5138218	
NDUFS5	NADH dehydrogenase (ubiquinone) Fe-S protein 5, 15kDa (NADH-coenzyme Q reductase)	2	rs10888650	6.30 × 10^−5^	2.90 × 10^−2^	−13.27	1	39041489	[8,9]
VN1R84P	vomeronasal 1 receptor 84 pseudogene	1	rs2015481	8.31 × 10^−5^	3.61 × 10^−2^	13.59	19	21676192	
LOC101928775	uncharacterized LOC101928775	1	rs10982832	8.33 × 10^−5^	3.61 × 10^−2^	16.77	9	116000000	
FGF19	fibroblast growth factor 19	1	rs7105655	1.01 × 10^−4^	4.24 × 10^−2^	13.69	11	69738836	
CARD17	caspase recruitment domain family, member 17	3	rs1623342	1.09 × 10^−4^	4.42 × 10^−2^	15.14	11	105000000	[7,9]
MIR4527	microRNA 4527	1	rs982265	1.17 × 10^−4^	4.67 × 10^−2^	−16.51	18	47444531	
Y_RNA	Y RNA	1	rs2248978	1.18 × 10^−4^	4.69 × 10^−2^	13.64	12	105000000	

Highlighted in light yellow are the genes that were not found in the previous studies.

**Table 3 cells-12-00578-t003:** The 16 placental disease-specific eQTL.

Gene Symbol	Description	Chr	Strand	Group	Keyword	References
*ERICH1*	glutamate rich 1	chr8	-	Multiple_Complex	PE, IUGR	[32] Identified as eGene by [9]
*DNAJC15*	DnaJ (Hsp40) homolog, subfamily C, member 15	chr13	+	Multiple_Complex	PE	Identified as eGene by [8,9]
*TAS2R64P*	taste receptor, type 2, member 64, pseudogene	chr12	-	Multiple_Complex		
*NSF*	N-ethylmaleimide-sensitive factor	chr17	+	Multiple_Complex	Membrane fusion	[33] Identified as eGene by [7,9]
*NPM1P35*	nucleophosmin 1 (nucleolar phosphoprotein B23, numatrin) pseudogene 35	chr11	+	Pseudogene	NPM1P35	
*PTTG1*	pituitary tumor-transforming 1	chr5	+	Multiple_Complex	Trophoblast invasion	[34] Identified as eGene by [9]
*C8orf89*	chromosome 8 open reading frame 89	chr8	-	Multiple_Complex	C8orf89	
*LINC00654*	long intergenic non-protein coding RNA 654	chr20	-	NonCoding	LINC00654	
*LINC01273*	long intergenic non-protein coding RNA 1273	chr20	+	NonCoding	LINC01273	
*FGF12-AS1*	FGF12 antisense RNA 1	chr3	+	NonCoding	tumor suppressor	[35]
*MYRF*	myelin regulatory factor	chr11	+	Multiple_Complex	autism	[36]
*ITPKB-IT1*	ITPKB intronic transcript 1	chr1	-	NonCoding		
*CITF22-49E9.3*	novel transcript	chr22	-	NonCoding		
*SLC30A8*	solute carrier family 30 (zinc transporter), member 8	chr8	+	Multiple_Complex	gestational weight gain, diabetes	[37]
*PROSER2-AS1*	PROSER2 antisense RNA 1	chr10	-	NonCoding	Placental imprinted, risk for pediatric fracture,	[38]
*PKN3*	protein kinase N3	chr9	+	Multiple_Complex	endothelial cell activation, angiogenesis	[39]

-/+ stands for the reference strand of DNA on which the gene is situated in the coding orientation.

**Table 4 cells-12-00578-t004:** **Best-eSNP-eGene** pairs for the MINUS DISEASE dataset showing the summary statistics from the eQTL analysis and the results of testing the significance of the interaction between genotype and disease on gene expression. The Best-eSNP, in this case, is defined as the most significant cis-eQTL associated with each eGene (smallest recorded *q*-value). Beta corresponds to the slope coefficient for the multivariate linear model and is an estimation of the eSNP effect size on gene expression variance. The table shows the test *p*-values for each coefficient of the linear model to test the effect of genotype and disease on gene expression and their interaction, as well as the model *p*-value. eGenes that presented a statistically significant interaction between eSNP genotype and disease status (*p*-value ≤ 0.05) are highlighted in green.

		From MatrixEQTL Analysis MINUS DISEASE			From Linear Regression to Test Genotype-Disease Interaction				
eGene	Best-eSNP	*p*-Value	FDR	beta	Intercept *p*-Val	eSNP *p*-Val	Group *p*-Val	eSNP*Group *p*-Val	Model *p*-Val
ZSCAN9	rs1150707	1.50 × 10^−16^	3.39 × 10^−12^	18.807	0.102647651	0.022415409	0.00787265	0.007064547	2.70 × 10^−11^
ERAP2	rs2549778	1.27 × 10^−12^	6.63 × 10^−9^	18.894	0.237759766	0.000807833	0.029435982	0.310364841	2.02 × 10^−9^
TOB2P1	rs13408	1.38 × 10^−12^	6.63 × 10^−9^	−17.608	0.004647281	0.001326778	0.886480707	0.50710562	1.53 × 10^−8^
PSG7	rs7248225	2.47 × 10^−12^	1.03 × 10^−8^	26.757	0.01610927	0.000161224	0.644681149	0.935717136	1.80 × 10^−8^
AQP11	rs10793257	1.10 × 10^−11^	3.37 × 10^−8^	20.693	0.936757201	0.244954435	0.002232625	0.009747518	3.07 × 10^−8^
RP1-97J1.2	rs9372316	1.30 × 10^−10^	3.42 × 10^−7^	21.469	0.04677023	0.02282663	0.98324046	0.5861815	1.34 × 10^−5^
LOC646029	rs10793257	1.59 × 10^−10^	3.91 × 10^−7^	20.487	0.62483721	0.449090131	0.000702104	0.003472929	2.96 × 10^−8^
RPS26	rs11171739	3.39 × 10^−10^	8.00 × 10^−7^	18.460	0.05049322	0.04455226	0.32794464	0.08493583	3.95 × 10^−7^
IL36RN	rs6761276	2.54 × 10^−9^	5.23 × 10^−6^	−17.273	0.04374326	0.00261548	0.37625035	0.56160599	3.05 × 10^−7^
CASP1P2	rs1023954	3.22 × 10^−8^	4.76 × 10^−5^	19.003	0.08623347	0.01572294	0.91051179	0.45255771	2.25 × 10^−5^
LYNX1	rs10956986	3.60 × 10^−8^	4.86 × 10^−5^	17.143	0.077423009	0.005369633	0.261583699	0.381727504	6.46 × 10^−7^
ANAPC4	rs1993602	1.00 × 10^−7^	1.26 × 10^−4^	−16.357	0.62788117	0.22099995	0.06658666	0.10987541	4.00 × 10^−6^
MIR4804	rs2253215	1.90 × 10^−7^	2.15 × 10^−4^	22.803	0.73283414	0.05996773	0.08001675	0.27408907	6.05 × 10^−6^
ZFYVE19	rs10152371	2.32 × 10^−7^	2.54 × 10^−4^	16.012	0.29373204	0.05488449	0.18253459	0.21546712	4.07 × 10^−6^
AC104135.2	rs7573356	5.00 × 10^−7^	4.97 × 10^−4^	−20.261	0.1301216	0.146468	0.7380213	0.2503687	3.31 × 10^−4^
GBP3	rs12121223	8.00 × 10^−7^	7.81 × 10^−4^	−19.408	0.025397469	0.004030585	0.591077883	0.775339911	8.26 × 10^−6^
OR2F1	rs7798409	9.03 × 10^−7^	8.59 × 10^−4^	−20.163	0.2916771	0.1675941	0.5381669	0.4819719	1.89 × 10^−3^
FUT10	rs12155581	2.26 × 10^−6^	1.93 × 10^−3^	17.033	0.931411621	0.806002691	0.013310468	0.004690609	5.07 × 10^−6^
MIR3927	rs7046565	2.41 × 10^−6^	2.04 × 10^−3^	16.732	0.020431665	0.002434153	0.405350891	0.294832332	5.07 × 10^−4^
GPR132	rs7157567	3.86 × 10^−6^	2.96 × 10^−3^	18.710	0.06895758	0.03121023	0.83496731	0.76965232	2.64 × 10^−3^
ERICH1	rs1703911	5.35 × 10^−6^	3.90 × 10^−3^	−13.888	0.048146347	0.003527824	0.9634337	0.940283908	3.92 × 10^−5^
DNAJC15	rs17553284	6.26 × 10^−6^	4.43 × 10^−3^	−19.484	0.469141	0.2646227	0.0112012	0.0594694	2.67 × 10^−5^
AGA	rs4690523	9.97 × 10^−6^	6.57 × 10^−3^	−16.364	0.50371342	0.14900309	0.06567993	0.18234469	7.27 × 10^−5^
TAS2R64P	rs7297949	1.05 × 10^−5^	6.85 × 10^−3^	−16.077	0.0824359	0.02563944	0.86790276	0.878125	2.53 × 10^−4^
AC009542.2	rs10954493	1.07 × 10^−5^	6.95 × 10^−3^	−18.232	0.97990037	0.03550566	0.05021882	0.21802336	7.21 × 10^−6^
GTSF1	rs11170917	1.46 × 10^−5^	8.84 × 10^−3^	22.336	0.34080682	0.02176511	0.67946143	0.83764288	2.02 × 10^−3^
CBLB	rs1503920	1.50 × 10^−5^	9.02 × 10^−3^	14.714	0.02302123	0.04722184	0.47428316	0.98231439	1.98 × 10^−3^
SOHLH2	rs9575600	1.65 × 10^−5^	9.76 × 10^−3^	16.140	0.10582542	0.01653465	0.87063264	0.92481755	3.25 × 10^−4^
KANSL1-AS1	rs17585974	1.67 × 10^−5^	9.81 × 10^−3^	23.860	0.11809498	0.06863156	0.62204626	0.77192668	4.91 × 10^−3^
NSUN7	rs2437317	1.73 × 10^−5^	9.94 × 10^−3^	23.457	0.01246184	0.01516865	0.08784461	0.56754666	3.00 × 10^−3^
NSF	rs17698176	1.74 × 10^−5^	9.97 × 10^−3^	20.367	0.22639048	0.00414694	0.7413922	0.79027276	4.09 × 10^−5^
NDUFS5	rs10888650	1.82 × 10^−5^	1.01 × 10^−2^	−13.620	0.52728811	0.42320815	0.24332657	0.03161536	8.82 × 10^−6^
CASQ1	rs6693877	1.98 × 10^−5^	1.10 × 10^−2^	−14.309	0.1020241	0.0981154	0.9077578	0.3968772	6.41 × 10^−4^
NPM1P35	rs4488202	2.38 × 10^−5^	1.29 × 10^−2^	−21.404	0.6814926	0.2694253	0.1968538	0.1925592	1.02 × 10^−3^
PTTG1	rs1105789	2.43 × 10^−5^	1.32 × 10^−2^	15.471	0.0267482	0.01723734	0.54238551	0.79645133	7.37 × 10^−4^
C8orf46	rs12541098	4.07 × 10^−5^	1.98 × 10^−2^	−15.621	0.01999623	0.01458403	0.4383046	0.67636289	2.10 × 10^−3^
C8orf89	rs6472746	4.65 × 10^−5^	2.19 × 10^−2^	−17.620	0.004634949	0.000225028	0.080493823	0.022123942	3.17 × 10^−4^
LINC00654	rs6116828	4.93 × 10^−5^	2.29 × 10^−2^	12.740	0.16132241	0.02684875	0.56544041	0.76958675	4.19 × 10^−4^
MTPAP	rs1762598	5.02 × 10^−5^	2.31 × 10^−2^	−15.032	0.05145106	0.01354065	0.49610048	0.4799675	4.82 × 10^−3^
CARD17	rs1623342	5.17 × 10^−5^	2.36 × 10^−2^	15.497	0.30416008	0.03915456	0.60188393	0.84825756	1.63 × 10^−3^
SLC13A5	rs9889374	5.49 × 10^−5^	2.48 × 10^−2^	15.823	0.10733714	0.53938538	0.84592422	0.05661093	4.53 × 10^−4^
LINC01273	rs6020255	5.91 × 10^−5^	2.66 × 10^−2^	−14.134	0.4831266	0.461141	0.2441706	0.1435129	1.01 × 10^−3^
LOC101928775	rs10982832	5.95 × 10^−5^	2.66 × 10^−2^	16.227	0.19619467	0.01188732	0.8784099	0.98711196	1.54 × 10^−4^
VN1R84P	rs2015481	6.03 × 10^−5^	2.68 × 10^−2^	13.299	0.01188751	0.10670774	0.3118514	0.42971681	2.99 × 10^−4^
CPS1	rs2250976	7.35 × 10^−5^	3.09 × 10^−2^	14.858	0.19187507	0.09947661	0.68926108	0.80062325	7.05 × 10^−3^
FGF12-AS1	rs10937543	8.44 × 10^−5^	3.44 × 10^−2^	−17.045	0.53243677	0.07383697	0.22123134	0.52782373	4.57 × 10^−4^
MYRF	rs7925523	1.02 × 10^−4^	4.12 × 10^−2^	18.305	0.049161643	0.000625781	0.393915774	0.080254068	5.05 × 10^−4^
ITPKB-IT1	rs697845	1.03 × 10^−4^	4.15 × 10^−2^	19.480	0.006181248	0.000344546	0.057691247	0.056088085	5.11 × 10^−4^
MIR4299	rs7126296	1.04 × 10^−4^	4.15 × 10^−2^	15.096	0.03207948	0.01109053	0.62800608	0.46298706	3.04 × 10^−3^
CITF22-49E9.3	rs137878	1.12 × 10^−4^	4.38 × 10^−2^	−17.668	0.20391064	0.02392672	0.9198819	0.67454661	4.21 × 10^−3^
SLC30A8	rs10505312	1.18 × 10^−4^	4.57 × 10^−2^	20.222	0.7540111	0.457554	0.3942176	0.4400821	3.34 × 10^−2^
CTC-482H14.5	rs2620833	1.25 × 10^−4^	4.74 × 10^−2^	16.078	0.084645173	0.008758073	0.852309458	0.531835913	1.04 × 10^−3^
PROSER2-AS1	rs7900122	1.26 × 10^−4^	4.75 × 10^−2^	−15.424	0.1660911	0.03140426	0.69555964	0.72078802	1.49 × 10^−2^
PKN3	rs10819449	1.32 × 10^−4^	4.91 × 10^−2^	19.043	0.3171833	0.416917	0.9639612	0.2775942	9.58 × 10^−3^

## Data Availability

The significant data of the study are provided in detail in the Appendix A. Acess to the complete dataset can be obtained on request from the authors.

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
