# Peer review of "Pan-Genomic Regulation of Gene Expression in Normal and Pathological Human Placentas"

_cells, 2023, doi:10.3390/cells12040578_

Round 1
Reviewer 1 Report
The study attempts to find genetic variants that affect gene expression (eQTL) in human (presumably delivered, though this is never explicitly stated) placentas, both normal and pathological. However, the manuscript is poorly written, contains grammatical/punctuation errors, and employs a highly questionable method of analysis, which they indicate is the novelty of the research.
I have several fundamental problems with the manuscript. First, the analysis method does not reflect the intended causal direction. The model specifies the outcome to be gene expression and the exposure/covariate to be the disease status. However, this is not an interesting question -- whether gene expression is effected by gene status. Rather, the authors hope to identify potentially causal mechanisms FOR disease, not DUE to disease. Thus, the appropriate model would have used disease as the outcome (multinomial logistic regression) and expression as the primary covariate of interest. Furthermore, the inclusion of 23 covariates with only 57 samples is never warranted. Justification for the choice of covariates is not provided. The authors indicate an evaluation of confounding/covariates by use of correlation coefficients. However, this is not how confounding is evaluated. Rather, a change in effect size when the variable is included in the model >15% is the usual standard. Only variables significantly associated with the outcome or meet criteria for confounding should be included in the model. Collinearity was inappropriately assessed; VIFs should be used. Model fit was not assessed. Including GA as a covariate (as well as fetal sex) likely results in residual confounding; matching should have been performed instead. This should be discussed as a limitation. It does not appear that an expert statistician analyzed this data. As a result of these major deficiencies, I must regretfully recommend rejection of this manuscript.
Specific Comments:
Abstract:
1. It is unclear to me -- and this relates to the the analysis in general -- why the covariate for the disease status was ever removed. I do not understand the purpose of this analysis.
2. Overstatement that 16 transcripts IDd de-novo must be related to placental disease.
Introduction:
1. The first sentence is extremely difficult to comprehend the author's intended thought.
2. Line 43: grammar mistake "humans evolves" should be humans evolve.
3. Line 45: "and so on" is too informal
4. There are many more articles on this topic than the author sites or alludes to.
5. "Bioinformatics tools" is vague.
6. Lines 62 - 92 are either methods or results, but should not be included in the introduction.
7. Line 68: It is unclear what was used as the outcome here. Sounds like it's gene expression levels, which implies that the disease is leading to the changes in expression level, when what you want is the other way around.
8. Line 71-72: There is no justification, nor reason to think, that applied statistical analysis would suggest that n=57 is sufficient and no power analysis is presented.
9. Line 73-74: What is meant by "are modified"? Again, conceptually, the disease status is the outcome, resulting from the alterations in gene expression. It is unclear why the investigators would model "reverse causation."
10. Line 77: Unfounded statement. In no way can the authors include that observed results will reflect causes, rather than effects. They can merely hypothesize, based on biology.
11. Unclear why the measure of residual variance is used to assess eQTL. This should be further explained and referenced. The choice of 0.85 should also be justified.
12. Lines 89-92: Adding covariates to the model will decrease the power, in most instances. There is no justification for this statement.
13. Figure 1 is terribly confusing and thus not very helpful.
14. There are virtually no methods described -- placenta collection, processing, statistical analysis, from where and whom samples were obtained, etc.
Results -- please note that an extremely thorough assessment was not made since the methods used for analysis were not consistent with the hypothesis, that these eQTL are causal.
15. Not extremely clear (except in abstract) that "normal" placentas were also collected. Not described who is considered "normal."
16. Table 1: Report frequency along with counts. Unknown sample sizes for each variable. Was no data missing for any variable?
17. Line 111 -- "removal of covariates affecting gene expression" is unclear and makes little sense as written. Again reminding the authors that they included far too many covariates with the available N. Should have included no more than ~1-3, carefully chosen due to being predictors or confounders (properly defined).
18. Line 115: Variance of what?
19: Figure 2: Inappropriate way to assess confounding.
20. Correlation coefficient cutoff for collinearity unusually high. Also should have calculated VIFs to assess it.
21. Line 136: I am assuming cohort means disease status? That's unclear. Again, this method of analysis (with expression as the outcome of the disease) is inconsistent with not only the proposed causal model, but is also not interesting. We can imagine that disease state DOES alter gene expression and by using delivered (affected or not) placentas makes this a more likely causal model for this experiment. However, it is not of interest to observe change DUE to disease, but changes that are along the causal pathway TO the disease.
22. Lines 152-161: I am not familar with the use of residual variance as a way to reduce the number of tested genes. This does not make sense, given what I attempted to find on the topic in the literature.
Please see A Statistical Framework for Joint eQTL Analysis in Multiple Tissues
Timothée Flutre,
Xiaoquan Wen,
Jonathan Pritchard,
Matthew Stephens
23. Lines 169-170: Again, This should be justified according to what is generally accepted in the literature. It should also be indicated whether this value was chosen a priori or based on data.
24. Line 228-229: There is insufficient power for testing an interaction. Futhermore, this was never listed as one of the aims of the study, and is thus not aligned with the hypothesis.
[Remaining Results were not heavily scrutinized, as explained above.]
Overarching remaining concerns about manuscript:
The interpretation of the results should not be included in the results section, but rather, the discussion section, which appears to be entirely absent from this manuscript. There is no discussion of limitations or strengths of the study. Statistical methods appear to be included in the results section but should be included in its own section in methods.
Author Response
There are three native English speakers in the authors. As for the first version, the last one was accurately re-read. If some errors remain, we hope that they are benign and could be corrected when the paper is edited.
Please see the detailed response to the different points in the attached file.

Reviewer 2 Report
The authors investigated pan-genomic regulation of gene expression in normal and pathological human placentas. They attempted to find genetic variants affecting gene expression in the placentas, in normal and pathological situations. They used a bioinformatics workflow that accounted for discrepancies in gestational age and other confounding variables in normal vs disease placentas. They found a highly significant overlap with previous studies, but also identified novel gene candidates that may have a role in placental biology. The approach may also be applied to the study of other human diseases where confounding factors have hampered a better understanding of the pathology.
Overall, this was a clearly presented and well conducted study and will be of interest to both placental biologists as well as others in settings where confounding factors have hampered a better understanding of pathology.
Minor points:
Table 3 should be presented correctly (formatting).
Author Response
We thank the reviewer for the openness of her/his vision, we improved the presentation of Table 3 and reposition it inside the text.
Round 2
Reviewer 1 Report
This study attempts to identify gene variants that impact gene expression in the human placenta, comparing normal to PE and IUGR. Overall, it is scientifically sound and of moderate interest to reproductive researchers. One limitation is the use of delivered placentas (not specified if these were all vaginal or some c-section), as these may well be reflecting the disease process rather than etiologic factors. Thus, reverse causality should be discussed as a limitation.
Overall:
1. Not adjusting for fetal sex in gene expression studies is rather problematic
2. Use of delivered placentas limits information on causal direction and should be acknowledged.
3. Detailed information as to how placentas were collected, processed, sampled, etc. is needed. Note that mode of delivery, formalin fixation, etc. can impact gene expression.
4. Detailed information on control placentas not provided. Who served as controls?
5. Large number of covariates included in model with n=57 not clearly justified. The most parsimonious model should be chosen and selection of covariates requires justification.
6. Authors seem unclear as to what a confounder is -- is NOT assessed by correlation coefficients.
Specific Comments:
Abstract:
1. Fetal sex is an important factor in gene expression. Since reports of differences in sex according to PE status, it is an important covariate to consider in analysis of placental gene expression.
2. Statistical correction for GA is not ideal, as when examining PE, adjusting may lead to collider bias. Can a GA-matched analysis be performed?
3. Line 28: Conclusion that if related to placental gene expression must therefore be related to placental disease is too strong a statement and should be "softened."
Introduction:
1. Lines 35-37: Unclear as to what the authors are saying.
Line 43: Grammar: humans evolve
Line 45: Delete "and so on."
Lines 52-53: There are more than 3 papers that have examined the "landscape of genetically controlled genes in the placenta."
Line 60: Clarify sentence; vague
Lines 62-92: These are methods, not introduction
Lines 65-68: Unclear which is the dependent variable. Sounds like gene expression was the "outcome" and disease state was considered the independent variable, however, this does not make sense biologically as a causal model would then suggest that the disease "causes" gene expression, when the reverse is what the authors appear to want. Please clarify independent and dependent variables.
Lines 71-72: Reference for "robust strategy" and limited sample size claim.
Line 77: Unfounded statement when evaluating delivered placentas, post dz-state.
Lines 89-92: Reference this claim, if possible.
Section 2.3:
1. I am not familiar with the use of residual variance as a way to reduce the number of tested genes. This is not consistent with what I see in the literature.
Please see A Statistical Framework for Joint eQTL Analysis in Multiple Tissues
Timothée Flutre,
Xiaoquan Wen,
Jonathan Pritchard,
Matthew Stephens
2. Lines 169-170: Please justify choice of threshold in relation to what is supported in the literature.
Note: Results are adequately described, though clarity could be improved.
Discussion:
1. GA is strongly associated with disease status as a result of the disease. When looking at the relationship between disease status and gene expression, GA is clearly associated with gene expression, but the association between disease and GA is not that GA is a risk (or protective) factor for the disease and therefore, adjusting for GA can lead to collider bias. This presents a rather tricky issue since we know that GA does strongly impact gene expression, but adjusting can also be problematic. This should be discussed, at a minimum.
2. No strengths/limitations of the study are identified. Please consider the various strengths and limitations as well as the possible impact of those limitations on results.
Author Response
Please find also our response in the PDF document.
Comments and Suggestions for Authors
This study attempts to identify gene variants that impact gene expression in the human placenta, comparing normal to PE and IUGR. Overall, it is scientifically sound and of moderate interest to reproductive researchers. One limitation is the use of delivered placentas (not specified if these were all vaginal or some c-section), as these may well be reflecting the disease process rather than etiologic factors. Thus, reverse causality should be discussed as a limitation.
Overall:
- Not adjusting for fetal sex in gene expression studies is rather problematic
Thank you for this remark; this limitation is mentioned in a novel Fifth part of our paper (Limits and conclusions) lines 672-674.
- Use of delivered placentas limits information on causal direction and should be acknowledged.
Sure, but this is very classical in almost all the placental gene expression studies. Anyway, our approach using the delivery as a covariate disqualifies its influence on the result. Besides, we are not searching for genetic causes of delivery. Since the diseases analysed (preeclampsia and IUGR) occur much before delivery (preeclampsia from the 20th week until the term, and IUGR is by definition intra-uterine). This question is nevertheless mentioned lines 675-676
- Detailed information as to how placentas were collected, processed, sampled, etc. is needed. Note that mode of delivery, formalin fixation, etc. can impact gene expression.
The technics used are described in the material and methods as much as generally done in the field. The details given in Material and Methods are standard procedures. We nevertheless added some information lines 143-145
- Detailed information on control placentas not provided. Who served as controls?
Classical controls in the field are just placentas obtained from uncomplicated pregnancies, neither for the mother, nor for the baby. This is mentioned line 125
- Large number of covariates included in model with n=57 not clearly justified. The most parsimonious model should be chosen and selection of covariates requires justification.
Yes you are right, it is not parsimonious, but as explained line 670-671, our choice was to reduce the Type I error, find reliable eSNPs. We think that the objective was reached since we confirmed a substantial part of the previous findings in the domain. We are aware that we may miss some hits, but we privileged the robustness of our findings.
- Authors seem unclear as to what a confounder is -- is NOT assessed by correlation coefficients.
In terms of gene expression, we observed that the part of the variability included in the Batch effect (the individual microarray experiment), or in the cohort (Paris, Angers or St George’s) represent more than 20% of the variance of the genes. These are factors that we do not want to influence our results, since they are not representing universal scientific questions. The other parameters (such as the clinical ones, sex, gestational age, maternal age, gestational age parity, and so on), might be research subjects and could be considered rather as covariates than as confounding factors, and with this we agree with the reviewer. Nevertheless, the aim of our approach was to be able to assess the influence of disease and genetics and distinguish these from all the parameters available taken as ‘confounders’. To try to clarify, we explain that we consider confounders as a large inclusive term as mentioned in the text lines 327-329.
Specific Comments:
Abstract:
- Fetal sex is an important factor in gene expression. Since reports of differences in sex according to PE status, it is an important covariate to consider in analysis of placental gene expression.
Thank you for this remark, it is true that sex accounts for 5% of the variance (line 337). Nevertheless, this question of links between sex and preeclampsia would be better addressed with a larger sample size.
- Statistical correction for GA is not ideal, as when examining PE, adjusting may lead to collider bias. Can a GA-matched analysis be performed?
A priori no, strictly speaking, preeclampsias are really very seldom going to term, and control placenta have no reason to be obtained at early terms. This is a systematic conundrum in the field. We estimate that our approach subtracting the GA effects might be the best available way around this issue.
- Line 28: Conclusion that if related to placental gene expression must therefore be related to placental disease is too strong a statement and should be "softened."
The sentence is now: “Then, we performed the analysis again removing the disease status from the covariates, and we identified 54 e-genes, 16 of which are identified de-novo, and thus, possibly related to placental disease.”
Introduction:
- Lines 35-37: Unclear as to what the authors are saying.
This has been improved in the previous correction run.
Line 43: Grammar: humans evolve
This was corrected line 47
Line 45: Delete "and so on."
This was done previously
Lines 52-53: There are more than 3 papers that have examined the "landscape of genetically controlled genes in the placenta."
True if you do not strictly speak about eQTL. This has been explicated lines 55-59
Line 60: Clarify sentence; vague
We tried to improve it in the revision
Lines 62-92: These are methods, not introduction
This part was moved in the methods section. Now, the introduction ends line 81
Lines 65-68: Unclear which is the dependent variable. Sounds like gene expression was the "outcome" and disease state was considered the independent variable, however, this does not make sense biologically as a causal model would then suggest that the disease "causes" gene expression, when the reverse is what the authors appear to want. Please clarify independent and dependent variables.
Sorry we tried to explain this in our previous revision. In the field of placental diseases, there is no idea of causality between gene expression and disease. The two directions are possible; the disease can trigger changes in gene expression and it does for sure (that may depend in their intensity of the genetic variation inside these genes), but reciprocally, gene expression alterations caused by a anterior cause (such as altered immunological dialogue between the uterus and the early placenta) could lead to pathological pregnancies. In our work there is no attempt to reach causality. To give an example, defects of implantation caused by improper immunological dialogue very early (around 9 days after fertilization), may trigger defective implantation and defective invasion by the extravillous trophoblast, that will later trigger alterations of placental vascularization, which will lead to oxidative/nitrosative stress, which is a major component of preeclampsia, which will lead to necrotize parts of the placenta, which will increase the oxidative/nitrosative stress, which will aggravate the symptoms and so on. Speaking about causality in such a cascade is an impregnable challenge. What we assume is that gene variants can make genes in specific cascades more or less responsive to the environmental conditions, and aggravate or alleviate the disease outcome or its advent. This point of non-causality is evoked lines 75-77.
Lines 71-72: Reference for "robust strategy" and limited sample size claim.
You are right. ‘robust’ was removed line 100
Line 77: Unfounded statement when evaluating delivered placentas, post dz-state.
Sorry, the line does not correspond to the last version of the text.
Lines 89-92: Reference this claim, if possible.
This is what our text says: ‘In order to restrict the number of input genes for the analysis, we first built a linear model using the gene expression levels for each gene in function of the covariates, then we selected a subset of potentially relevant genes based on their residual variance (the variability of their expression level) since genes for which there is no residual variance once the covariate effects are removed, are not going to reveal any detectable effect that could be partly explained by changes at the genetic level. This approach has similarly be used in a QTL research paper for genomic methylation QTL (meQTL, Bergesdt et al., 2022).’ So Bergesdt et al is a good reference to the methodology
Section 2.3:
- I am not familiar with the use of residual variance as a way to reduce the number of tested genes. This is not consistent with what I see in the literature.
Please see A Statistical Framework for Joint eQTL Analysis in Multiple Tissues
Timothée Flutre,
Xiaoquan Wen,
Jonathan Pritchard,
Matthew Stephens
Bergesdt et al is also a very good reference for this approach. What you propose is for multiple tissue, we study only the placenta.
- Lines 169-170: Please justify choice of threshold in relation to what is supported in the literature.
I suppose you make reference to the 0.85 threshold. We added a justification based upon linkage disequilibrium consideration and the genotyping array density lines 395-396 ‘At this threshold, at least 6 SNP per gene were significant, meaning that the association with gene expression is not merely due to linkage disequilibrium’
Note: Results are adequately described, though clarity could be improved.
Discussion:
- GA is strongly associated with disease status as a result of the disease. When looking at the relationship between disease status and gene expression, GA is clearly associated with gene expression, but the association between disease and GA is not that GA is a risk (or protective) factor for the disease and therefore, adjusting for GA can lead to collider bias. This presents a rather tricky issue since we know that GA does strongly impact gene expression, but adjusting can also be problematic. This should be discussed, at a minimum.
We now discuss this in the limitations (lines 681-683)
- No strengths/limitations of the study are identified. Please consider the various strengths and limitations as well as the possible impact of those limitations on results.
We mention the question as saying that our work is preliminary (‘a starting point for other researchers’).
